# Dose- and Sex-Dependent Changes in Hemoglobin Oxygen Affinity by the Micronutrient 5-Hydroxymethylfurfural and α-Ketoglutaric Acid

**DOI:** 10.3390/nu13103448

**Published:** 2021-09-29

**Authors:** Simon Woyke, Norbert Mair, Astrid Ortner, Thomas Haller, Marco Ronzani, Christopher Rugg, Mathias Ströhle, Reinhold Wintersteiger, Hannes Gatterer

**Affiliations:** 1Department of Anaesthesiology and Critical Care Medicine, Medical University of Innsbruck, Anichstraße 35, 6020 Innsbruck, Austria; simon.woyke@i-med.ac.at (S.W.); marco.ronzani@tirol-kliniken.at (M.R.); christopher.rugg@tirol-kliniken.at (C.R.); 2Institute of Physiology, Medical University of Innsbruck, Schöpfstraße 41, 6020 Innsbruck, Austria; norbert.mair@i-med.ac.at (N.M.); thomas.haller@i-med.ac.at (T.H.); 3Institute of Pharmaceutical Sciences, University of Graz, Schubertstraße 1, 8010 Graz, Austria; astrid.ortner@uni-graz.at (A.O.); reinhold.wintersteiger@uni-graz.at (R.W.); 4Institute of Mountain Emergency Medicine, Eurac Research, Via Ipazia 2, 39100 Bolzano, Italy; hannes.gatterer@eurac.edu

**Keywords:** 5-HMF, αKG, oxygen dissociation curve, P50, Hill coefficient

## Abstract

5-Hydroxymethylfurfural (5-HMF) is known to increase hemoglobin oxygen affinity (Hb–O_2_ affinity) and to induce a left shift of the oxygen dissociation curve (ODC). It is under investigation as a therapeutic agent in sickle cell anemia and in conditions where pulmonary oxygen uptake is deteriorated or limited (e.g., various clinical conditions or altitude exposure). The combination of 5-HMF and α-ketoglutaric acid (αKG) is commercially available as a nutritional supplement. To further elucidate dose effects, ODCs were measured in vitro in venous whole blood samples of 20 healthy volunteers (10 female and 10 male) after the addition of three different doses of 5-HMF, αKG and the combination of both. Linear regression analysis revealed a strong dose-dependent increase in Hb–O_2_ affinity for 5-HMF (R^2^ = 0.887; *p* < 0.001) and the commercially available combination with αKG (R^2^ = 0.882; *p* < 0.001). αKG alone increased Hb–O_2_ affinity as well but to a lower extent. Both the combination (5-HMF + αKG) and 5-HMF alone exerted different P50 and Hill coefficient responses overall and between sexes, with more pronounced effects in females. With increasing Hb–O_2_ affinity, the sigmoidal shape of the ODC was better preserved by the combination of 5-HMF and αKG than by 5-HMF alone. Concerning the therapeutic effects of 5-HMF, this study emphasizes the importance of adequate dosing in various physiological and clinical conditions, where a left-shifted ODC might be beneficial. By preserving the sigmoidal shape of the ODC, the combination of 5-HMF and αKG at low (both sexes) and medium (males only) doses might be able to better maintain efficient oxygen transport, particularly by mitigating potentially deteriorated oxygen unloading in the tissue. However, expanding knowledge on the interaction between 5-HMF and Hb–O_2_ affinity in vitro necessitates further investigations in vivo to additionally assess pharmacokinetic mechanisms.

## 1. Introduction

The hemoglobin oxygen dissociation curve (ODC) characterizes the reversible binding of four molecules of oxygen to hemoglobin (Hb) [1]. The main parameters describing the ODC include the oxygen partial pressure at half saturation (P50) and the Hill coefficient (HC), with the latter depicting the maximum steepness of the ODC in the Hill plot and thus the cooperativity of the oxygen binding to Hb [2]. The ODC is affected by several known and unknown factors [3], which either increase Hb–O_2_ affinity, shifting the ODC to the left; or decrease Hb–O_2_ affinity, shifting it to the right [1]. Whether one or the other may be beneficial in various disease states or environmental conditions is currently under debate [4], yet the targeted modulation of Hb–O_2_ affinity might be a future therapeutic target [5,6].

The micronutrient 5-HMF increases Hb–O_2_ affinity by way of allosteric modification of the Hb molecule by creating a Schiff-base adduct after transversing the red blood cell membrane [7,8]. In a hamster trial, 5-HMF increased Hb–O_2_ affinity, prevented hemodynamic instability and maintained microvascular oxygenation [7]. In pigs exposed to hypoxia, 5-HMF also increased Hb–O_2_ affinity, which led to an improved arterial oxygen saturation (SO_2_) and mitigated the hypoxia related increase in pulmonary arterial pressure [9]. Additionally, in healthy humans exposed to hypoxia as well as patients suffering from sickle cell anemia, Hb–O_2_ affinity was increased by 5-HMF [10,11]. Furthermore, the combination of 5-HMF and α-ketoglutaric acid (αKG), a commercially available nutritional supplement (Sanopal^®^), was shown to increase peripheral SO_2_ in healthy participants cycling at a simulated altitude of 3500 m (F_i_O_2_ = 13.5%) [12]. Despite being a strong anion and therefore hydrophilic, αKG is known to be rapidly transported across red blood cell membranes, where it can potentially affect Hb–O_2_ affinity [13,14]. A prospective randomized trial in lung cancer patients showed positive effects of 5-HMF and αKG (Sanopal^®^) supplementation on maximum oxygen consumption (VO_2_max) and hospitalization time after single lung ventilation during thoracic surgery [15].

These three conditions—namely hypoxia, sickle cell anemia, and lobectomy with single lung ventilation—share a possible benefit from increased Hb–O_2_ affinity. It is intriguing to speculate that not only in these conditions but also for different patient groups affected by systemic hypoxemia (e.g., acute respiratory distress syndrome, chronic obstructive pulmonary disease or COVID-19), increasing Hb–O_2_ affinity (e.g., with 5-HMF administration) may be a beneficial intervention. Accordingly, it has been suggested that the modification of Hb by pharmacological agents or supplements may have positive effects on COVID-19 patients [6,16,17]. As mentioned previously, the nutritional supplement Sanopal^®^, which contains a specific combination of 5-HMF and αKG, modifies Hb–O_2_ affinity. However, there is still lack of knowledge about the effects of different 5-HMF and αKG doses and their combination on potential ODC shifts, which is an essential prerequisite for the assessment of possible clinical relevance.

Thus, this study aimed to investigate whether 5-HMF or αKG changes Hb–O_2_ affinity in a dose dependent manner in human whole blood, and particularly whether it changes the cooperativity of the Hb–O_2_ binding, as indicated by the HC. Additionally, the combined effect of 5-HMF and αKG on the ODC was analyzed. A balanced representation of male and female subjects allowed the investigation of possible sex effects.

## 2. Materials and Methods

This study was approved by the ethics committee of the Medical University of Innsbruck (vote nr. 1265/2020) and registered with ClinicalTrials.gov (NCT04612270). Written informed consent was obtained from all subjects.

Venous blood was collected from 20 healthy volunteers (10 female, 10 male) aged 18 to 40 years. Subjects were nonsmokers, not pregnant or breastfeeding and had no known hemoglobinopathy nor recent history of illness, trauma, blood loss or multi-day trips to high altitude (>3000 m). Immediately after blood sampling, a blood gas analysis was performed (ABL800 flex, Radiometer, Denmark). Samples were stored on ice and processed within eight hours. 5-HMF and αKG were dissolved in different carrier solutions (a = aqua; b = glucose 30 g/L; c = glucose and phosphoric acid at pH 5.7; d = aqua and phosphoric acid at pH 5.7). Control solutions consisted of the sole carrier without 5-HMF or αKG, respectively. In controls, 10 µL of sole carrier solution was added to 100 µL of whole blood. In the treatment groups, 10 µL of each of three different concentrations of 5-HMF, αKG or 5-HMF plus αKG were added to 100 µL of whole blood and gently mixed by resuspension. The maximum recommended daily 5-HMF dose of Sanopal^®^ (CYL Health GmbH, Pöllau bei Hartberg, Austria) is 720 mg in total. Assuming equal distribution in a 70 kg person’s blood volume of about 5000 mL [18], the low dose of 5-HMF was defined as 0.14 mg/mL. Medium dose (0.6 mg/mL, 3000 mg/70 kg) was chosen to correspond to a common dosage of Aes-103, a 5-HMF containing anti-sickle agent [10]. High dose (2 mg/mL, 10,000 mg/70 kg) corresponds to a dosage of 5-HMF (143 mg/kg) that showed adverse events in animal trials, such as a rise in serum gamma-globulin levels and relative spleen weight [19,20]. The αKG was dosed to reach a 3:1 ratio with 5-HMF. 

We used a novel in-vitro high-throughput method for the recording of ODCs, which is based on a modified, gas-perfused 96-well microplate and a fluorescence microplate reader [21]. Duplicate measurements were performed for every concentration in the thin films of unbuffered whole blood samples. For the ODC experiments gas mixtures containing 40 mmHg PCO_2_ were used, and the temperature was held constant at 37 °C. 

Curve fittings as well as P50 and HC calculations were performed and graphics were generated using Excel (2016, Microsoft, Redmond, WA, USA), statistical analysis was done using IBM SPSS Statistics (v25, IBM Corp, New York, NY, USA). Unpaired t-test was used to analyze differences between carrier solution compositions in controls (whole blood sample plus carrier solution without 5-HMF or αKG). ANOVA with repeated measurement design was used to detect P50 and HC differences between the different doses and substance compositions. Post hoc t-test (Bonferroni corrected) and ANOVA with repeated measurement design for each single substance or a combination of two substances were used to identify the location and the degree of the changes and to compare the effect of sex. Linear regression analysis was used to determine concentration dependence for each substance and between sexes. *p* < 0.05 was considered significant. Data are presented as mean ± standard deviation (SD).

## 3. Results

The mean age of all subjects was 29.6 ± 3.0 years. Hb concentrations (14.5 ± 1.3 g/dL) and hematocrit (Hct; 44.4 ± 4.0%) were within normal ranges, while pH was slightly lower than 7.40 (7.35 ± 0.04) corresponding to higher carbon dioxide levels (PCO_2_; 48.4 ± 7.4 mmHg) in venous blood samples (see Table 1 for sex differences). In controls, P50 was 25.1 ± 1.3 mmHg, showing a slight sex difference (females 26.0 ± 1.0 mmHg vs. males 24.3 ± 0.9 mmHg; *p* = 0.001) and HC values were 2.61 ± 0.21.

As the statistical analysis showed no differences with regard to carrier solution (Figure 1), results of different carrier solutions per concentration level were averaged for the final analysis.

ANOVA repeated measurement design showed an overall substance (*p* < 0.001), dose (*p* < 0.001) and interaction (substances × dose, *p* < 0.001) effect for both, P50 and HC (Figure 2a,b). Results for P50 and HC are shown in Table 2.

5-HMF significantly decreased P50 (*p* < 0.001; Figure 2a and Figure 3a) and HC (*p* < 0.001; Figure 2b) in a dose-dependent manner. The αKG alone also modified P50 (*p* < 0.001) and HC (*p* = 0.012), but to a much lesser extent compared to 5-HMF alone (Figure 2; Figure 3c). The combination (5-HMF + αKG) showed an overall dose and substance effect regarding P50 and HC (*p* < 0.001). Compared to 5-HMF alone, differences in P50 were particularly present at the high combined dosage (*p* = 0.028) (Figure 2a), whereas differences in HC were more distinct at low and medium doses (*p* < 0.001 and *p* = 0.015) (Figure 2b). Sex differences are described below. All analyzed substances and combinations showed a linear correlation between dose and effect on P50 (Figure 3).

Analyses showed an interaction effect of dose and sex on P50 for 5-HMF + αKG and 5-HMF alone (5-HMF: *p* < 0.001; 5-HMF + αKG: *p* = 0.043; αKG: *p* = 0.397; Figure 2a). Similar effects were found for the HC (5-HMF: *p* = 0.001; 5-HMF + αKG: *p* = 0.012; αKG: *p* = 0.292; Figure 2b). In females, a more pronounced 5-HMF effect on P50 using medium to high doses was shown (Figure 2a). Interestingly, the medium dose of 5-HMF decreased P50 to the same level as the combination with αKG in females, while in men the medium dose of 5-HMF alone led to a higher increase in Hb–O_2_ affinity than the combination. Similarly in both sexes, the high dose of combined 5-HMF and αKG potentiated the effect of 5-HMF alone.

Overall, changes in P50 and HC affected the total shape of the ODC in specific ways (Figure 4). At low doses of 5-HMF and 5-HMF + αKG and for the control solution, the typical sigmoidal shape of the ODC was clearly visible. At medium doses, the sigmoidal character was still preserved, but due to the higher HC (when averaging females and males), the ODC for 5-HMF + αKG compared to 5-HMF alone showed an even better preserved sigmoidal shape. At high doses, both ODCs lost their typical sigmoidal shape due to very low P50 and very low HC values.

## 4. Discussion

The main findings of this study are that 5-HMF and αKG increased Hb–O_2_ affinity by decreasing both P50 and the HC in a dose-dependent manner in human whole blood. The analyzed substances alone and in combination exerted an influence on the ODC, yet to different degrees (i.e., αKG showed the least effect). Interestingly, the combination of 5-HMF and αKG at low and medium doses decreased the HC to a lesser extent than 5-HMF alone, therefore better preserving the sigmoidal shape of the ODC (Figure 4).

The dose-dependent influence of 5-HMF on the ODC was already hypothesized by Stern et al. who reported changes in peripheral SO_2_ at simulated altitude after ingesting different doses of the 5-HMF containing substance “Aes-103”, an anti-sickling agent [10]. The present study extends these findings by showing a linear dose effect of 5-HMF on the measured ODC. The question arises as to what the adequate dose and thus extent of ODC modification may be to induce clinically desirable changes. Kössler et al. showed a small but significant increase in peripheral SO_2_ (1.3% peripheral SO_2_ differences) during exercise at altitude in subjects taking the combination of 5-HMF and αKG, compared to those in the placebo group [12]. Matzi et al. administered 720 mg 5-HMF daily for 10 days before lung surgery in lung cancer patients [15]. They reported an improvement in VO_2_max and a reduction in hospitalization time in the treatment group. Hb–O_2_ affinity was not measured, but since pulmonary oxygen uptake is critical in these patients, that is, in single lung ventilation during surgery, it may be suggested that the increased Hb–O_2_ affinity (due to the supplementation) and the resulting SaO_2_ increase might have contributed to the reported benefits [15].

It must be clearly stated that a leftward shift of the ODC has the potential to impair peripheral O_2_ unloading, which is critical in most diseases. Therefore, enhancing pulmonary oxygen uptake by therapeutically increasing Hb–O_2_ affinity may potentially deteriorate oxygen delivery at the tissue level [6,20]. Particularly in patients suffering from coronary heart disease, ischemic stroke or peripheral vascular disease, this issue should be rigorously considered and should be addressed in further studies. It is important to consider that there may be only a few conditions that could benefit from a left-shifted ODC, for instance hypoxemia, sickle cell anemia, single lung ventilation, acute respiratory distress syndrome, COPD, or Covid-19.

The results of this study can be seen as the basis for such considerations at least if 5-HMF is the agent of choice. Considering the linear dose characteristics of 5-HMF and αKG, low to medium doses seem to have a sufficient effect depending on the desired outcome. Doses chosen too high, on the other hand, may induce massive left shifts with concomitant disappearance of the sigmoidal shape of the ODC, which is unlikely to be beneficial in clinical practice.

An interesting finding of the present investigation is the combined effect of 5-HMF and αKG on the HC. Changes in the HC indicate an influence on the sigmoidal shape of the ODC [2,22], which is critical for Hb oxygen transport. In general, a steeper slope of the ODC supports Hb oxygen transport [2] by facilitating O_2_ loading in the lungs and O_2_ unloading at the tissue level, thus maximizing oxygen exploitation. Therefore, the adverse effect of a high Hb–O_2_ affinity on oxygen delivery to the tissue might be mitigated by a steep slope of the ODC, expressed by a high HC. At low and medium doses, the combination of 5-HMF with αKG, in comparison to 5-HMF alone, seemed to preserve the sigmoidal shape of the ODC while increasing Hb–O_2_ affinity (Figure 4). Whether such small differences may have physiological relevance is unclear, as are the reasons why the addition of αKG modifies the HC. In regard to the latter, it could be suggested that by exhibiting substantial anti-oxidative capacity (hence avoiding excessive oxidative stress), undesirable protein–protein interactions which may influence Hb cooperativity, may be avoided [23]. For the high dose of 5-HMF and the combination with αKG, HC barely exceeded 1.0, indicating that the sigmoidal character of the ODC, ensuring the steep slope of the curve, disappeared. It is therefore doubtful that this dose would benefit hemoglobin oxygen transport.

Sex differences in Hb–O_2_ affinity have been reported and our data on the control samples support these previous findings [24,25]. However, differences between sexes in the effect of 5-HMF on Hb–O_2_ affinity have not been reported before. Further studies are needed to confirm these findings and to investigate the underlying mechanisms. As sex differences in the concentration of Hb (*p* < 0.001 for this data) are widely recognized, future studies should focus on sex specific differences in Hb functionality, that is Hb–O_2_ affinity.

Comparison of the used dosages with orally administrated Sanopal^®^ or Aes-103 need to be taken with caution as 100% bioavailability was assumed, which is unlikely the case with oral administration. In this study, dosages were chosen that would correspond to an immediate onset of drug effectiveness. However, pharmacokinetics, in particular rapid degradation of 5-HMF, should be considered when transferring the results to an in vivo situation. Furthermore, an effect of possible confounders (hemolysis, MetHb, 2,3-BPG changes) during the ODC measurements cannot be completely excluded, but can be considered unlikely as all groups including controls were treated equally and spiked with carrier solution. It has to be acknowledged that changes in pH due to the addition of study drugs (5-HMF, αKG), were not assessed. However, an effect of 5-HMF is not expected as it is fully protonated at physiological pH (pKa > 12). By contrast, αKG is a strong anion at physiological pH (pKa < 3), and therefore most probably decreased the pH of whole blood samples (anion gap acidosis). However, a decrease in pH should lead to a right shift in the ODC, which is contrary to the observed left shift detected for αKG alone and for the combination of 5HMF and αKG. Although the exact mechanisms of the observed Hb–O_2_ affinity modulation remain unknown, we are confident that pH shifts were most probably not responsible for the recorded effects.

## 5. Conclusions

5-HMF induced a strong dose-dependent increase in Hb–O_2_ affinity. By maintaining a higher HC at lower P50, the combination of 5-HMF with αKG, in comparison to 5-HMF alone, preserved the sigmoidal shape of the ODC while increasing Hb–O_2_ affinity, thus maintaining efficient oxygen transport, that is, oxygen delivery to the tissue. Females compared to males showed a more pronounced effect of 5-HMF on Hb–O_2_ affinity.

## Figures and Tables

**Figure 1 nutrients-13-03448-f001:**
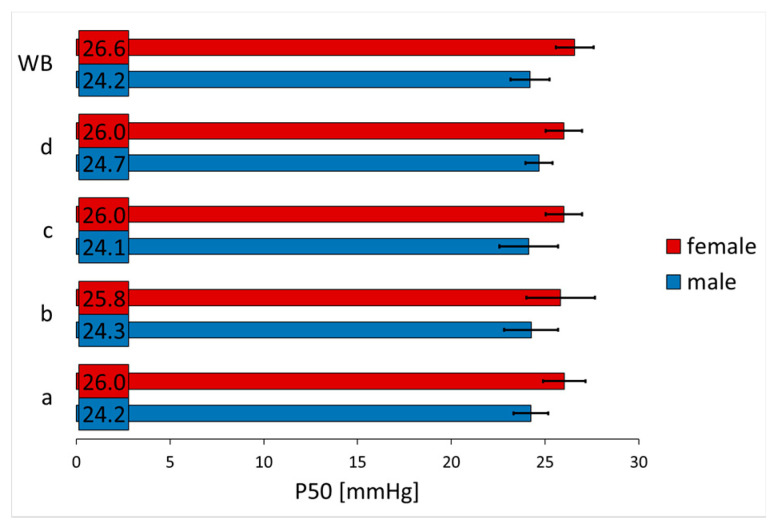
Oxygen partial pressure at half saturation (P50; mean ± SD) of whole blood (WB) and whole blood with 5-HMF- and αKG-free carrier solutions (controls; a = aqua; b = glucose 30 g/L; c = glucose and phosphoric acid at pH 5.7; d = aqua and phosphoric acid at pH 5.7).

**Figure 2 nutrients-13-03448-f002:**
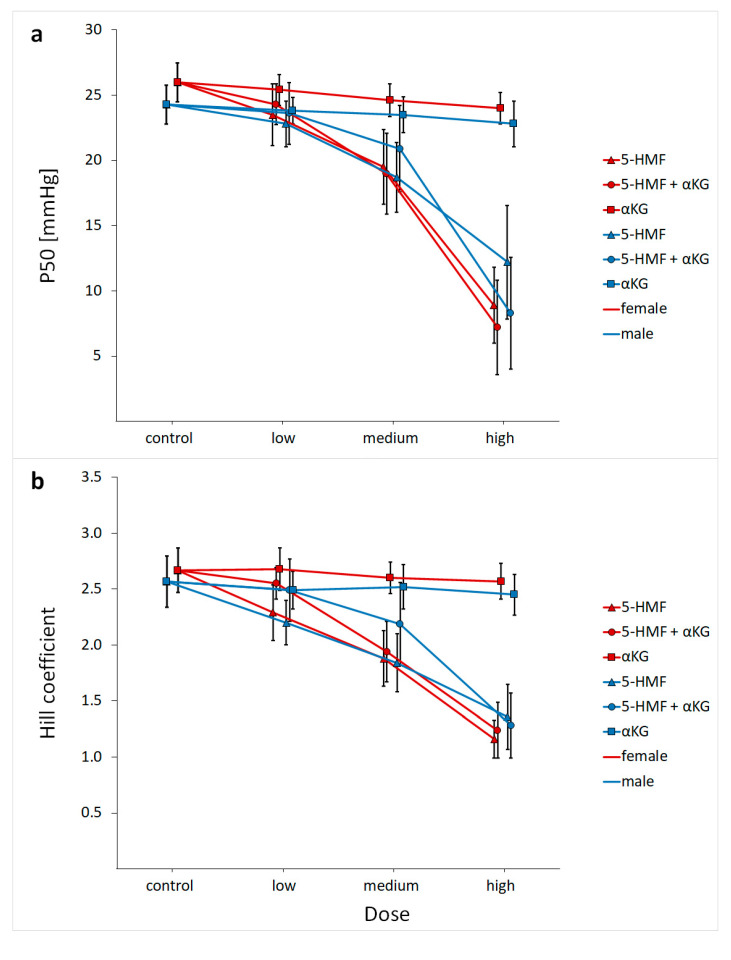
Effects of 5-hydroxymethylfurfural (5-HMF) and α-ketoglutaric acid (αKG) at different dose levels (5-HMF low dose = 0.14 mg/mL, 5-HMF medium dose = 0.6 mg/mL, 5-HMF high dose = 2 mg/mL; αKG low dose = 0.42 mg/mL, αKG medium dose = 1.8 mg/mL, αKG high dose = 6 mg/mL) on (**a**) oxygen partial pressure at half saturation (P50; mean ± SD) and (**b**) Hill coefficient (HC; mean ± SD).

**Figure 3 nutrients-13-03448-f003:**
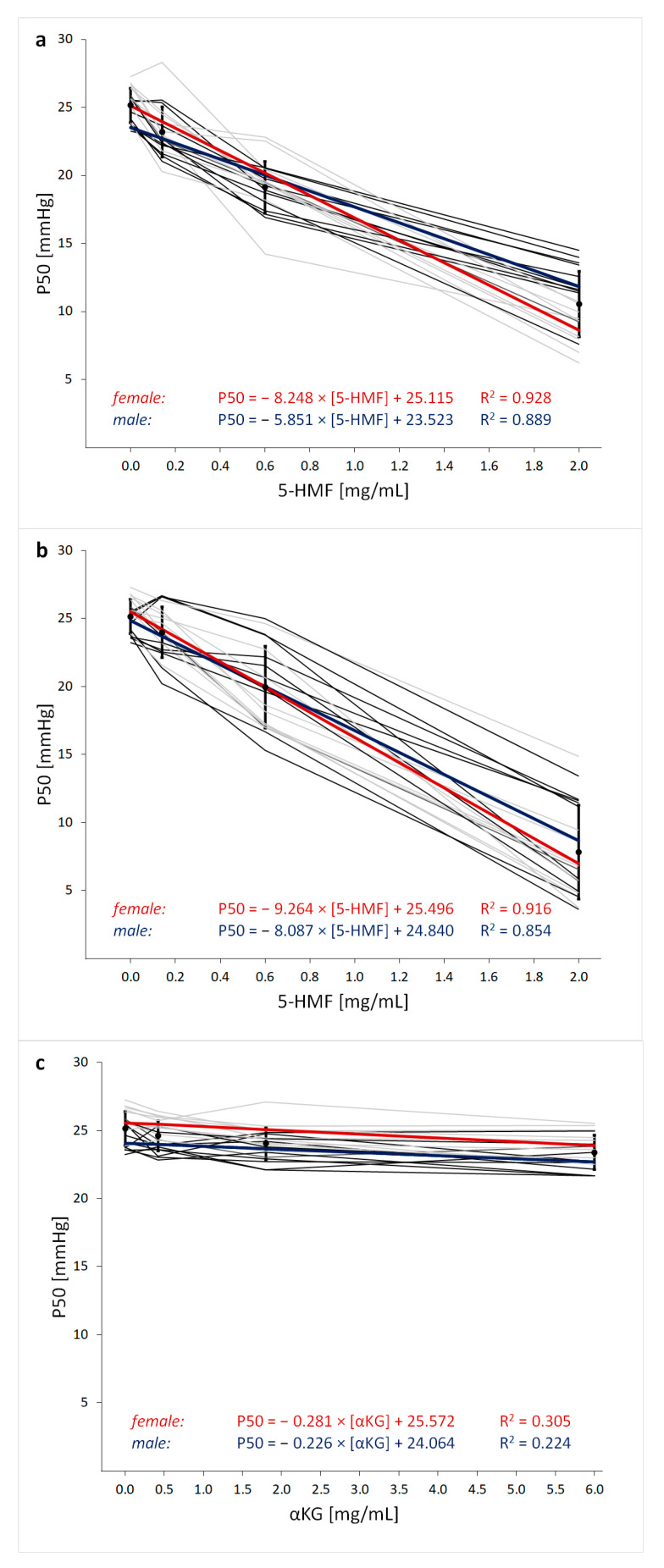
Dose-dependent changes in oxygen partial pressure at half saturation (P50) of (**a**) 5-hydroxy-methylfurfural (5-HMF) with α-ketoglutaric acid (αKG) (5-HMF + αKG), (**b**) 5-hydroxy-methylfurfural alone (5-HMF) and (**c**) α-ketoglutaric acid alone (αKG). Grey lines indicate individual courses for females and black lines for males, and black circles show the mean ± SD of all subjects. The colored lines represent the linear regression function with the parameters shown for females (red) and males (blue), respectively.

**Figure 4 nutrients-13-03448-f004:**
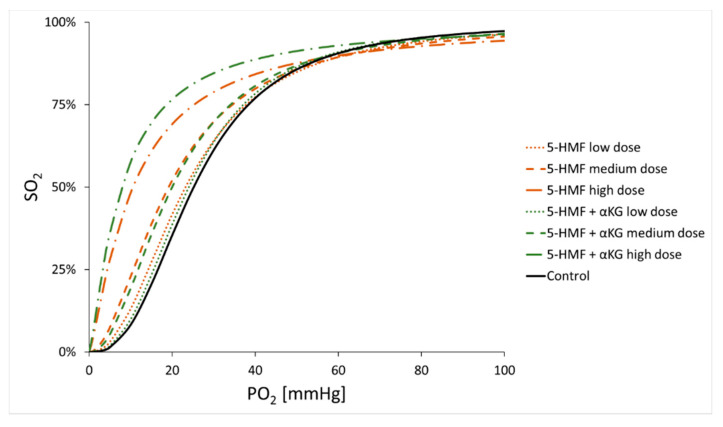
Comparison of oxygen dissociation curves (ODCs) of 5-hydroxymethylfurfural (5-HMF; orange) alone and 5-hydroxymethylfurfural (5-HMF) with α-ketoglutaric acid (5-HMF + αKG; green) as indicated by the interaction of mean oxygen partial pressure at half saturation (P50) and mean Hill coefficient (HC) of all subjects.

**Table 1 nutrients-13-03448-t001:** Characteristics of venous blood samples of subjects for both sexes (mean ± SD).

	Age (Years)	Hb (g/dL)	Hct (%)	pH	PCO_2_ (mmHg)
male	30.3 ± 4.0	15.5 ± 0.9	47.6 ± 2.6	7.35 ± 0.03	51.0 ± 6.7
female	28.9 ± 1.4	13.4 ± 0.7	41.3 ± 2.1	7.35 ± 0.04	45.8 ± 7.4
*p*	0.31	<0.001	<0.001	0.76	0.12

**Table 2 nutrients-13-03448-t002:** Oxygen partial pressure at half saturation (P50 in mmHg; mean ± SD) and Hill coefficient (HC; mean ± SD) for three doses of 5-hydroxymethylfurfural (5-HMF), α-ketoglutaric acid (αKG), and the combination of both for males (*n* = 10) and females (*n* = 10).

		P50 (mmHg)		HC	
		Females	Males	Females	Males
	control	26.0 ± 1.5	24.3 ± 1.5	2.7 ± 0.2	2.6 ± 0.2
5-HMF	low dose	23.5 ± 2.3	22.8 ± 1.7	2.3 ± 0.3	2.2 ± 0.2
	medium dose	19.5 ± 2.9	18.7 ± 2.7	1.9 ± 0.3	1.8 ± 0.3
	high dose	8.9 ± 2.9	12.2 ± 4.3	1.2 ± 0.2	1.4 ± 0.3
	low dose	24.3 ± 1.6	23.6 ± 2.4	2.6 ± 0.1	2.5 ± 0.3
5-HMF + αKG	medium dose	19.0 ± 3.1	20.9 ± 3.3	1.9 ± 0.3	2.2 ± 0.4
	high dose	7.2 ± 3.6	8.3 ± 4.3	1.2 ± 0.3	1.3 ± 0.3
	low dose	25.4 ± 1.2	23.8 ± 1.0	2.7 ± 0.2	2.5 ± 0.2
αKG	medium dose	24.6 ± 1.3	23.5 ± 1.4	2.6 ± 0.1	2.5 ± 0.2
	high dose	24.0 ± 1.2	22.8 ± 1.7	2.6 ± 0.2	2.4 ± 0.2

## Data Availability

The data that support the findings of this study are available from the corresponding author (MS) upon reasonable request.

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
