# Peer review of "Dose- and Sex-Dependent Changes in Hemoglobin Oxygen Affinity by the Micronutrient 5-Hydroxymethylfurfural and α-Ketoglutaric Acid"

_nutrients, 2021, doi:10.3390/nu13103448_

Round 1

Reviewer 1 Report

The aim of the study was to show dose and sex dependent changes in hemoglobin oxygen affinity by the micronutrients 5-hydroxymethylfurfural and α-ketoglutaric acid. 5-Hydroxymethylfurfural (5-HMF) is known to increase hemoglobin oxygen affinity and to induce a left shift of the oxygen dissociation curve (ODC). The combination of 5-HMF and α-ketoglutaric acid (αKG) was shown to increase peripheral SO2 in healthy participants cycling at a simulated altitude of 3500 m.

In venous whole blood samples of 20 healthy volunteers (10 female and 10 male) ODCs were measured in vitro after addition of three different doses of 5-HMF, αKG  and the combination of both. The study revealed a strong dose-dependent increase in Hb-O2 affinity for 5-HMF and the commercially available combination with  αKG. αKG alone increased Hb-O2 affinity, too, but to a lower extent. Both the combination (5-HMF + αKG) and 5-HMF alone showed more pronounced effects in females. The sigmoidal shape of the ODC was better preserved by the combination of 5-HMF and αKG than by 5-HMF alone what contributes to maintain efficient oxygen transport in terms of oxygen unloading in the tissue.

The study was performed with a clear design, meaningful statistical analysis tools and adequate graphic illustration of the results. The discussion of the results is well done and the limitations of the study are pointed out, especially the limited transferability of the results to an in vivo study design. Unfortunately the authors did not define their terms “low dose”, “medium dose” and “high dose” for the both agents used in the study. Only by measuring within the graphs the dosages in mg/ml can be traced back for 5-HMF and αKG as well.

The results of the study are most interesting for treatment of patients suffering from different kinds of lung diseases as well as for the prevention of severe oxygenation problems in high altitude. The treatment of patients suffering from emphysema or chronic obstructive pulmonary disease (COPD) may be improved by the application of 5-HMF and αKG. The main finding of the present study, that 5-HMF and αKG increase Hb-O2 affinity in a linear dose dependent manner in human whole blood, will help to design prospective in vivo studies to prove the described effects under natural conditions.

Author Response

REVIEWER 1:

The aim of the study was to show dose and sex dependent changes in hemoglobin oxygen affinity by the micronutrients 5-hydroxymethylfurfural and α-ketoglutaric acid. 5-Hydroxymethylfurfural (5-HMF) is known to increase hemoglobin oxygen affinity and to induce a left shift of the oxygen dissociation curve (ODC). The combination of 5-HMF and α-ketoglutaric acid (αKG) was shown to increase peripheral SO2 in healthy participants cycling at a simulated altitude of 3500 m.

In venous whole blood samples of 20 healthy volunteers (10 female and 10 male) ODCs were measured in vitro after addition of three different doses of 5-HMF, αKG and the combination of both. The study revealed a strong dose-dependent increase in Hb-O2 affinity for 5-HMF and the commercially available combination with αKG. αKG alone increased Hb-O2 affinity, too, but to a lower extent. Both the combination (5-HMF + αKG) and 5-HMF alone showed more pronounced effects in females. The sigmoidal shape of the ODC was better preserved by the combination of 5-HMF and αKG than by 5-HMF alone what contributes to maintain efficient oxygen transport in terms of oxygen unloading in the tissue.

The study was performed with a clear design, meaningful statistical analysis tools and adequate graphic illustration of the results. The discussion of the results is well done and the limitations of the study are pointed out, especially the limited transferability of the results to an in vivo study design. Unfortunately the authors did not define their terms “low dose”, “medium dose” and “high dose” for the both agents used in the study. Only by measuring within the graphs the dosages in mg/ml can be traced back for 5-HMF and αKG as well.

Dear Reviewer,

We would like to dearly thank you for your time and your favorable review!

With regard to the mentioned dosages, we must admit that they were not stated clear enough. We would like to apologize for this mistake.

Besides already being described within the Material and Methods section (page 3), we chose to also enclose numeric values for the given dosages within the legend of Figure 2. We now truly believe that readability and comprehensibility is increased. Thank you for this valuable hint!

The results of the study are most interesting for treatment of patients suffering from different kinds of lung diseases as well as for the prevention of severe oxygenation problems in high altitude. The treatment of patients suffering from emphysema or chronic obstructive pulmonary disease (COPD) may be improved by the application of 5-HMF and αKG. The main finding of the present study, that 5-HMF and αKG increase Hb-O2 affinity in a linear dose dependent manner in human whole blood, will help to design prospective in vivo studies to prove the described effects under natural conditions.

We are glad to hear that our manuscript meets your approval. We do also believe that prospective in vivo studies are required to further confirm our findings.

Thank you for your review!

Reviewer 2 Report

The Authors tested 5HMF and aKG as modulators of the Hb-O2 affinity in whole blood of female and male subjects. They found that 5HMF increases the Hb-O2 affinity but are unable to explain the underlying mechanisms. Although predominantly observational and preliminary, the manuscript may be potentially interesting, but several technical and substantial aspects must be addressed.

The Authors run the ODC in blood samples from 20 subjects in the presence of 5HMF or aKG or both at different dosages but did not address the question whether the RBC membrane is permeable to these substances and can therefore have direct effects on the Hb-O2 binding.

As the Authors based their experiments on a newly developed method to measure the ODC that still needs full validation, they need to exclude the presence of critical factors that may affect the Hb-O2 affinity. Although the publication that first reported the method excluded pH variations during the ODC run, in this application they must exclude the possibility that the addition of 5HMF or aKG to the RBC suspension alters the pH, which is known to have enormous impact on the Hb-O2 affinity. The Authors must also account for possible changes during the ODC run (and therefore on the Hill coefficient) of the degree of RBC hemolysis and of metHb formation. Also 2,3-DPG may be a confounding variable, although perhaps to a lesser extent that those cited above. These uncertainties may reflect in an abnormally low normal P50, about 25 mmHg, with respect to normal standards.

Figure 1 shows the effect of carrier solution on the P50 without apparent effects, but data in Figure 2 must be compared at each dosage with vehicle-only data and must report SD/SEM bars. Perhaps multiple-way ANOVA would be the most appropriate statistical test to address the differences among sex and dosages.

The Authors must clearly point out if increased Hb-O2 affinity is beneficial (line 60) or not (line 202). If what matters more is the artero-venous difference, then the second option appears more appealing, and the statement in line 28 (abstract) is totally wrong: the O2 unloading to tissues is LESS efficient with increased Hb-O2 affinity. In such a case, 5HMF is not beneficial for humans in vivo and the rationale for performing this study becomes obscure. To speculate on the in vivo positive effects of 5HMF, the Authors should show pharmacokinetic data, with particular concern of the stability and the persistence of 5HMF and aKG in the circulation. Note that if these substances can enter the RBC and can thus have direct effects on the Hb-O2 binding, then their persistence in the blood stream becomes questionable.

The Authors must launch some ideas on how aKG and 5HMF can modify the Hill plot and the P50, and why males and females react differently to either.

Minor

Line 46, the Hb-O2 affinity is increased under all the conditions, not only in “severe hypoxia”.

Author Response

REVIEWER 2:

The Authors tested 5HMF and aKG as modulators of the Hb-O2 affinity in whole blood of female and male subjects. They found that 5HMF increases the Hb-O2 affinity but are unable to explain the underlying mechanisms. Although predominantly observational and preliminary, the manuscript may be potentially interesting, but several technical and substantial aspects must be addressed.

The Authors run the ODC in blood samples from 20 subjects in the presence of 5HMF or aKG or both at different dosages but did not address the question whether the RBC membrane is permeable to these substances and can therefore have direct effects on the Hb-O2 binding.

Dear Reviewer,

We must truly thank you for your time and the precise reading of our manuscript. Your input is of very high value, and we are certain our manuscript has improved substantially by addressing the issues raised by you!

Regarding RBC membrane permeability, we agree that this issue was completely left out, leaving the reader in uncertainty. As there is literature on this topic, we extended the introduction with the needed information and added the following:

 Page 2 line 54-56: “The micronutrient 5-HMF increases Hb-O2 affinity via allosteric modification of the Hb molecule by creating a Schiff-base adduct after transversing the red blood cell membrane (8, 9).”

Page 2 line 67-70: “Despite being a strong anion and therefore hydrophilic, αKG is known to be rapidly transported across red blood cell membranes, where it can potentially affect Hb-O2 affinity (14, 15).“

As the Authors based their experiments on a newly developed method to measure the ODC that still needs full validation, they need to exclude the presence of critical factors that may affect the Hb-O2 affinity. Although the publication that first reported the method excluded pH variations during the ODC run, in this application they must exclude the possibility that the addition of 5HMF or aKG to the RBC suspension alters the pH, which is known to have enormous impact on the Hb-O2 affinity. The Authors must also account for possible changes during the ODC run (and therefore on the Hill coefficient) of the degree of RBC hemolysis and of metHb formation. Also 2,3-DPG may be a confounding variable, although perhaps to a lesser extent that those cited above. These uncertainties may reflect in an abnormally low normal P50, about 25 mmHg, with respect to normal standards.

Once more we must acknowledge the Reviewer for pointing out a major limitation of our study.

However, many concerns must be seen in the light that controls were as well spiked with the same amount and composition of carrier solution as were treatment groups. Figure 1 compares P50s of whole blood and whole blood mixed with different carrier solutions in a 1:10 ratio. No differences in P50 were detected leading to the assumption that the carrier solution does not alter pH (or other factors leading to ODC shifts). RBC hemolysis and MetHb formation, of course, may occur during any measurement or inadequate storage. However, we are confident that these changes would rather occur when using alternative methods like the Hemox Analyzer or other methods that are based on stirring or sparging. In this study 5-HMF and aKG whole blood samples are treated in the same way as controls, thus methodological issues regarding changes during the ODC run (hemolysis; metHb, 2,3-BPG) most probably would have affected all groups and therefore do not explain group differences.

We agree that 2,3-BPG is a main and important modifier of Hb-O2 affinity. Unfortunately, for over 2 years now, the enzymatic kit of sigma Aldrich is not available anymore. Several publications of other authors worldwide also criticize the impossibility to measure 2,3-BPG levels at the time. Different baseline 2,3-BPG levels are not expected within our study design, and changes in 2,3-BPG during the measurements are seen as unlikely as 2,3-BPG changes usually take a certain amount of time (hours to days) and the blood in this study was processed directly after taking the samples.

Unfortunately, we must fully agree that pH was not measured after adding the carrier solution AND the study drug to the whole blood samples. The effects of 5HMF and aKG on the acid-base-status of the sample were not assessed. However, 5-HMF has an approximated pKa > 12 and is therefore fully protonated at pH 7.4. Influences on sample pH are therefore not expected. aKG is a strong ion with a low pKa (<3) and therefore fully deprotonated at pH 7.4. Depending on dosage, the addition of this strong anion to whole blood samples is most likely to lead to a decrease in pH (strong ion gap positive acidosis).

In order to confirm this theory, we exemplary drew blood from two of our authors and measured pH via blood gas analysis before and after the addition of 5-HMF, aKG and the combination of both (corresponding to the highest dose used within the study). 5-HMF in the highest dose did not affect pH, aKG in its highest dose decreased pH in terms of an anion gap positive acidosis (min pH 7.2, BE -10, anion gap 70).  

However, a decrease in pH should lead to a right shift in the ODC, which is contrary to the observed left shift that was detected for aKG alone and for the combination of 5HMF and aKG.

Although the exact mechanisms of HbO2-affinity modulation remain unknown within our study, we must conclude that pH shifts were most probably not responsible for the observed effects.

We decided to extend the limitations section as follows:

Page 9 line 319ff: “Furthermore, the effect of possible confounders (hemolysis, MetHb, 2,3-BPG changes) during the ODC measurements cannot be completely excluded, but can be considered unlikely as all groups including controls were treated equally and spiked with carrier solution. It has to be acknowledged that changes in pH due to the addition of study drugs (5-HMF, αKG), were not assessed. However, an effect of 5-HMF is not expected as it is fully hydronated at physiological pH (pKa >12). In contrary, αKG is a strong anion at physiological pH (pKa <3), therefore most probably decreasing pH of whole blood samples (anion gap acidosis). However, a decrease in pH should lead to a right shift in the ODC, which is contrary to the observed left shift detected for αKG alone and for the combination of 5HMF and αKG. Although the exact mechanisms of the observed Hb-O2 affinity modulation remain unknown, we are confident that pH shifts were most probably not responsible for the recorded effects.”

Figure 1 shows the effect of carrier solution on the P50 without apparent effects, but data in Figure 2 must be compared at each dosage with vehicle-only data and must report SD/SEM bars. Perhaps multiple-way ANOVA would be the most appropriate statistical test to address the differences among sex and dosages.

Dear Reviewer,

we are sorry for not presenting the mentioned data clear enough. In fact, vehicle-only data are shown in Fig 2a and Fig 2b. They are declared as controls (positioned as first point on the x-axis). Comparison was conducted with regard to these controls, which represent whole blood samples with added carrier solution only. As each participants’ blood samples were measured as control and consecutively after adding 5-HMG, aKG or the combination, we are confident that ANOVA with repeated measurements is an appropriate approach. Of course, we must admit that a multiple-way ANOVA would be just as adequate. Furthermore, we fully agree with your issue regarding SD-bars and included these in the Figure 2. Mean and SD of all combinations are shown in table 2.

For better clarity we modified the corresponding section within the methods section:

Page 3 line 120ff:

“Control solutions consisted of the sole carrier without 5-HMF or αKG, respectively. In controls, 10 µl of the sole carrier solution was added to 100 µl of whole blood. In the treatment groups, 10 µl of each of three different concentrations of 5-HMF, αKG or 5-HMF plus αKG were added to 100 µl of whole blood and gently mixed by resuspension.”

The Authors must clearly point out if increased Hb-O2 affinity is beneficial (line 60) or not (line 202). If what matters more is the artero-venous difference, then the second option appears more appealing, and the statement in line 28 (abstract) is totally wrong: the O2 unloading to tissues is LESS efficient with increased Hb-O2 affinity. In such a case, 5HMF is not beneficial for humans in vivo and the rationale for performing this study becomes obscure. To speculate on the in vivo positive effects of 5HMF, the Authors should show pharmacokinetic data, with particular concern of the stability and the persistence of 5HMF and aKG in the circulation. Note that if these substances can enter the RBC and can thus have direct effects on the Hb-O2 binding, then their persistence in the blood stream becomes questionable.

Dear Reviewer,

Thank you for pointing out a very important potential misunderstanding in our study.

Regarding the mentioned sentence in the abstract, we completely agree that the previous version read potentially misunderstanding. What we meant was that the combination of 5HMF and aKG in medium and low doses, maintained a potentially better oxygen unloading in the tissue WHEN compared to 5HMF alone, by better preserving the sigmoidal shape. We rephrased the sentence to the following:

“By preserving the sigmoidal shape of the ODC, the combination of 5-HMF and αKG at low and medium doses might be able to better maintain efficient oxygen transport, particularly by mitigating potentially deteriorated oxygen unloading in the tissue”

Whether or not an increased Hb-O2 affinity in vivo is beneficial, highly depends on various factors and cannot be easily broken down to a simple yes or no answer. We completely agree with the Reviewer that an increased affinity potentially means an impaired oxygenation at the tissue level. Apart from many patient-intern factors, the amount of affinity increase and the preservation of the sigmoidal shape are of utmost importance regarding resulting beneficence or maleficence. In this regard, we do believe that this study provides high-quality information. The aim of this study was to provide in vitro dose-effect data of 5-HMF and aKG impact on Hb-O2 affinity. We strongly believe that the rationale of this study is not obscure by no means. In contrary, we believe that this is the first key step to enable scientists worldwide to perform in vivo studies, where clinically beneficial effects for patients can be further proven or falsified.

The Authors must launch some ideas on how aKG and 5HMF can modify the Hill plot and the P50, and why males and females react differently to either.

We fully agree that this issue is of great interest.

5-HMF is known to modify Hb by creating a “Schiff-base”. As mentioned before, we included a short statement in the introduction section.

Unfortunately, the mechanism how aKG modifies Hb is unclear and not the main objective of this study. As also mentioned within the introduction aKG is rapidly transported through the RBC membrane. The observed left shift in the ODC by aKG alone is not explained by potential pH shifts as mentioned within the limitations. Lastly, further research is required to fully answer this question.

Also, the mechanism of sex dependency is unclear and not easy to answer. In this regard, we would like to point out that the aim of this study was to show dose dependent changes in Hb-O2 affinity of 5-HMF and aKG, also presenting significant sex differences. The underlying mechanisms can only be speculated on, as the study was not conducted accordingly and literature is scarce or non-existent. To avoid excessive speculation, the discussion section is kept concise.

Although underlying mechanisms cannot be shown, we hope that the results of this study will encourage other scientists to conduct further research on this issue. Thus we believe to add a valuable contribution to this research field.

Minor

Line 46, the Hb-O2 affinity is increased under all the conditions, not only in “severe hypoxia”.

We agree and rephrased accordingly.

Thank you for your review!

References in the Reply

  1. Dempsey JA. With haemoglobin as with politics - should we shift right or left? J Physiol. 2020;598(8):1419-20.
  2. Kössler F, Mair L, Burtscher M, Gatterer H. 5-Hydroxymethylfurfural and α-ketoglutaric acid supplementation increases oxygen saturation during prolonged exercise in normobaric hypoxia. Int J Vitam Nutr Res. 2019:1-6.
  3. Stern, W, D M, JC M, J S, K G. A Phase 1, First-in-Man, Dose-Response Study of Aes-103 (5-HMF), an Anti-Sickling, Allosteric Modifier of Hemoglobin Oxygen Affinity in Healthy Norman Volunteers. Blood. 2012(120:3210).

Round 2

Reviewer 2 Report

The Authors have done a good job in answering my queries, but a few issues still need to be cleared out.

“What we meant was that the combination of 5HMF and aKG in medium and low doses, maintained a potentially better oxygen unloading in the tissue WHEN compared to 5HMF alone, by better preserving the sigmoidal shape.” I can’t agree. Figure 2 shows that the combination 5HMF+aKG and 5HMF alone are indistinguishable at every dose, both for the P50 and the Hill coefficient. Therefore, the statement “By preserving the sigmoidal shape of the ODC, the combination of 5-HMF and αKG at low and medium doses might be able to better maintain efficient oxygen transport, particularly by mitigating potentially deteriorated oxygen unloading in the tissue” is speculative and must be revised.

No doubt that “Whether or not an increased Hb-O2 affinity in vivo is beneficial, highly depends on various factors and cannot be easily broken down to a simple yes or no answer.” But the contexts that would require higher Hb-O2 affinity essentially are pulmonary diseases where Hb oxygenation is impaired (ARDS, COPD, emphysema, COVID-19 and perhaps asthma), as well as uncompensated high-altitude acclimatization (maybe, many physiological studies at altitude support the opposite view). Instead, coronary heart disease, ischemic stroke or peripheral vascular disease need to be assessed in a case-by-case fashion. In virtually all the other cases, the variable to be protected with care is peripheral tissue oxygenation, e.g., lower Hb-O2 affinity (of course, within a reasonable range). Therefore, without questioning the potential impact of this study, the rationale underlying the use of 5HNF and aKG is to be clarified and re-addressed targeting fewer diseases.

Line 249, replace “hydronated” with “protonated”.

Figure 2. Try to nudge slightly the data points to avoid overlap of the error bars. Some popular plotting programs like GraphPad or similar can do it.

Author Response

The Authors have done a good job in answering my queries, but a few issues still need to be cleared out.

Again, we thank you for the excellent and detailed review. We strongly believe this revision will strengthen conciseness and consistency of this paper.

“What we meant was that the combination of 5HMF and aKG in medium and low doses, maintained a potentially better oxygen unloading in the tissue WHEN compared to 5HMF alone, by better preserving the sigmoidal shape.” I can’t agree. Figure 2 shows that the combination 5HMF+aKG and 5HMF alone are indistinguishable at every dose, both for the P50 and the Hill coefficient. Therefore, the statement “By preserving the sigmoidal shape of the ODC, the combination of 5-HMF and αKG at low and medium doses might be able to better maintain efficient oxygen transport, particularly by mitigating potentially deteriorated oxygen unloading in the tissue” is speculative and must be revised.

In Figure 2b red (female) and blue (male) triangles indicate Hill coefficients for 5-HMF alone, while circles show HC for the combination of 5-HMF and aKG. At low dose both circles (female + male) are distinctly higher than triangles, indicating higher HC and thus “a more sigmoidal shape of the ODC”. We agree, that at medium dose this effect is only pronounced for males. To be more precise, we included “both sexes” after low dose and “males only” after medium dose in parenthesis.

Regarding statistics, this sentence was based on the fact that the post-hoc Bonferroni comparison for HC at medium dose between 5-HMF alone vs. the combination is significant (P=0.015; see line 153), when females and males are analyzed together. We would also like to refer to Figure 4, where the different shape is also visible. Obviously, we agree that the differences are small, and to take this into account we added the following sentence to the discussion section “Whether such small differences may have physiological relevance is unclear as are the reasons why addition of αKG modifies the HC.”

No doubt that “Whether or not an increased Hb-O2 affinity in vivo is beneficial, highly depends on various factors and cannot be easily broken down to a simple yes or no answer.” But the contexts that would require higher Hb-O2 affinity essentially are pulmonary diseases where Hb oxygenation is impaired (ARDS, COPD, emphysema, COVID-19 and perhaps asthma), as well as uncompensated high-altitude acclimatization (maybe, many physiological studies at altitude support the opposite view). Instead, coronary heart disease, ischemic stroke or peripheral vascular disease need to be assessed in a case-by-case fashion. In virtually all the other cases, the variable to be protected with care is peripheral tissue oxygenation, e.g., lower Hb-O2 affinity (of course, within a reasonable range). Therefore, without questioning the potential impact of this study, the rationale underlying the use of 5HNF and aKG is to be clarified and re-addressed targeting fewer diseases.

We completely agree! We believe that targeted medical conditions are already carefully selected, i.e. hypoxemia, sickle cell anemia and lobectomy under single lung ventilation. However, we now included other conditions where a left shift might be beneficial, to provide a better overview and highlighted the fact that a left-shifted ODC might be beneficial in only a few diseases in lines 213-216.

Line 249, replace “hydronated” with “protonated”.

Thank you, we replaced hydronated.

Figure 2. Try to nudge slightly the data points to avoid overlap of the error bars. Some popular plotting programs like GraphPad or similar can do it.

Thank you for the hint. Now, we also separated the substances at each dose level. Error bars are now clearly visible for each substance, sex at each dose level.